# Distinct Thalamic and Frontal Neuroanatomical Substrates in Children with Familial vs. Non-Familial Attention-Deficit/Hyperactivity Disorder (ADHD)

**DOI:** 10.3390/brainsci13010046

**Published:** 2022-12-26

**Authors:** Rahman Baboli, Meng Cao, Jeffery M. Halperin, Xiaobo Li

**Affiliations:** 1Department of Biomedical Engineering, New Jersey Institute of Technology, Newark, NJ 07102, USA; 2Graduate School of Biomedical Sciences, Rutgers University, Newark, NJ 07039, USA; 3Department of Psychology, Queens College, City University of New York, New York, NY 11367, USA; 4Department of Electrical and Computer Engineering, New Jersey Institute of Technology, Newark, NJ 07102, USA

**Keywords:** ADHD, magnetic resonance imaging (MRI), heterogeneity, familial ADHD, non-familial ADHD, neuroanatomy, ABCD dataset, thalamus, inferior frontal gyrus

## Abstract

Attention-deficit/hyperactivity disorder (ADHD) is a highly prevalent, inheritable, and heterogeneous neurodevelopmental disorder. Children with a family history of ADHD are at elevated risk of having ADHD and persisting its symptoms into adulthood. The objective of this study was to investigate the influence of having or not having positive family risk factor in the neuroanatomy of the brain in children with ADHD. Cortical thickness-, surface area-, and volume-based measures were extracted and compared in a total of 606 participants, including 132, 165, and 309 in groups of familial ADHD (ADHD-F), non-familial ADHD (ADHD-NF), and typically developed children, respectively. Compared to controls, ADHD probands showed significantly reduced gray matter surface area in the left cuneus. Among the ADHD subgroups, ADHD-F showed significantly increased gray matter volume in the right thalamus and significantly thinner cortical thickness in the right pars orbitalis. Among ADHD-F, an increased volume of the right thalamus was significantly correlated with a reduced DSM-oriented t-score for ADHD problems. The findings of this study may suggest that a positive family history of ADHD is associated with the structural abnormalities in the thalamus and inferior frontal gyrus; these anatomical abnormalities may significantly contribute to the emergence of ADHD symptoms.

## 1. Introduction

Attention-deficit/hyperactivity disorder (ADHD) is one of the most prevalent and persistent neurodevelopmental disorders that affects approximately 5% of children worldwide [1,2]. There are three different clinical presentations for ADHD including primarily inattentive, primarily impulsive–hyperactive, and combined, and the diagnosis is based on the presence of developmentally impaired symptoms of inattention and/or impulsivity–hyperactivity for at least 6 months in multiple settings, such as at school and home [3,4]. Several studies suggest that the familial heritability is one of the most significant risk factors among both biological and environmental risk factors [5,6,7,8]. Relative to those without a positive family history (non-familial ADHD), children with a positive family history (familial ADHD) are at an increased risk of having ADHD with a significantly higher risk for persisting ADHD symptoms into adulthood [5,9,10,11]. Community-based studies reported similar rates of ADHD in both familial and non-familial subgroups [12,13]. In addition, a few clinical studies reported more impaired executive function in children with a positive family history [14,15]. The behavioral problems and cognitive impairments that are associated with familial ADHD might be partially contributed by the familial heritability-related disruptions in neurodevelopment. However, neuroanatomical determinants and their pathogenesis/clinical/behavioral linkages associated with familial ADHD (ADHD-F) vs. non-familial ADHD (ADHD-NF) remain unknown. Understanding of such mechanisms could lead to the discovery of novel biomarkers for early preventions in children with positive family histories and novel targets for individualized intervention, such as neuroprotective agents, in children with ADHD [16,17].

Existing structural neuroimaging studies in ADHD have shown widely inconsistent results. Several studies reported a lower cortical thickness in the pars opercularis, fusiform, parahippocampal, precentral gyri, temporal pole, and cingulate cortex [18,19,20], and a delay in cortical thickening in the prefrontal and premotor cortices [21]. Another study reported a higher surface area in children with ADHD in the medial superior temporal cortex and dorsomedial frontal cortex [22]. Reduced total brain volume, reduced gray matter volume in the bilateral frontal lobe, right orbitofrontal cortex, right primary cortex, and anterior cingulate cortex, and reduced subcortical volume in the accumbens, amygdala, caudate, hippocampus, and putamen have also been demonstrated in children with ADHD [18,23,24,25]. In addition, functional MRI (fMRI) studies reported that children with ADHD had functional alterations in the prefrontal cortex, striatal, and parietal brain regions during attention processes [26,27]. The functional alterations associated with the striatum and fronto-striatal pathways have been suggested to link with deficits in the response inhibition function in ADHD [28,29], while the functional alterations in the inferior parietal, superior parietal, and lateral prefrontal regions were suggested to contribute to deficits in spatial working memory and spatial attention [30,31]. These existing studies suggest that structural underdevelopments and functional alterations associated with frontal, striatal, and parietal regions may significantly contribute to the dysfunctions in behavioral inhibition and working memory in children with ADHD [32,33,34,35,36].

While there is a large quantity of literature on neural correlation of ADHD symptoms in samples of children/adults with ADHD compared to healthy control groups, there are notably less neuroanatomical imaging studies focused on the familial risk factor that may represent biologically distinct subgroups of ADHD [37,38,39]. Smaller inferior, medial, and orbitofrontal gray matter volumes have been reported in ADHD patients and their unaffected siblings compared to controls [39,40]. Other neuroimaging and neuropsychological studies focusing on familial risk factors have observed significant deficits in executive function and decreased functional connectivity during the performance of go/no-go tasks in the fronto-striatal and superior frontal in ADHD patients and their unaffected first-degree relatives compared to healthy control subjects [15,37,38,41,42,43]. These studies suggest that some frequently observed brain and behavioral abnormalities associated with the frontal lobe in ADHD may have heritable patterns. However, without having a clinical control group of ADHD-NF, these studies could not identify a familial risk-specific neurobiological mechanism in ADHD.

In the present study, we proposed to investigate the neuroanatomical substrates and their clinical/behavioral linkages associated with ADHD-F vs. ADHD-NF in a large sample of three independent groups of children (i.e., ADHD-F, ADHD-NF, and controls). On the basis of findings from our existing neuroimaging and clinical studies [44,45,46,47,48,49,50,51,52] and the family studies [37,38,40,41,42,43,53], we hypothesize that structural alterations in the frontal lobe and related circuits in ADHD may have a strong heritable familial origin, which can be differentiated from that in ADHD patients without a parental history of the disorder, suggesting a more prominent environmental etiology. 

## 2. Materials and Methods

### 2.1. Participants

All subjects were obtained from the baseline data in the Adolescent Brain Cognitive Development (ABCD) Study Release 4.0, a large study that took part in a major collaboration between multiple sites across the United States. All participants in the ABCD study were recruited from schools with diverse demographic factors, with the initial recruitment resulting in 11,875 children. All the parents provided written informed consent and the informed assent was obtained from children. The data were downloaded from the National Institute of Mental Health Data Archive. 

Participants were included in this study if they were in age ranged 9–10 and had Intelligence Quotient (IQ) higher than 80. (Picture vocabulary task’s t-score from NIH Toolbox battery [54] has been used to estimate IQ.) Participants were excluded from this study if they had a history of diagnosed traumatic brain injury using Modified Ohio State University TBI Screen-Short Version [55,56] or any neurological disorders; had diagnosed Bipolar Disorder, Schizophrenia, Autism, Pervasive Developmental Disorder, or a chronic tic disorder; had a biological parent with a history of diagnosed Autism, Bipolar disorder, Schizophrenia, or any type of psychosis; had unknown family medical history from either of the biological parents; had history of substance use in the past three months; had chronic medical illness; were on a non-stimulant psychotropic medication within past three months. 

Recently validated and computerized Kiddie Schedule for Affective Disorder and Schizophrenia (KSADS-5) was used to assess the ADHD symptomatology in the ABCD study [57,58]. In this study, we used the DSM-5 criteria for the current presentations of ADHD, i.e., participants who endorsed at least 6 of the 9 inattentive symptoms were considered as ADHD inattentive presentation, at least 6 of the 9 hyperactive/impulsive symptoms as ADHD impulsive/hyperactivity presentation, and at least 6 of 9 in both symptom domains as ADHD combined presentation, from at least 2 settings. After applying inclusion, exclusion, and ADHD symptomatology criteria, participants were categorized into three different groups as following, typically developed children (TDC) group, ADHD-NF group, and ADHD-F group. The ADHD-F group included those children who endorsed at least one of the ADHD presentations and had at least one biological parent that had previous diagnosis of ADHD or had significant inattentive and/or hyperactive–impulsive symptoms measured by t-scores > 65 in at least one of the three DSM-5 ADHD-oriented scales in the Adult Self-Report (ASR) [59]. The ADHD-NF group included those who endorsed at least one of the ADHD presentations and had no biological parent with previous/current diagnosis of ADHD or had inattentive or hyperactive–impulsive symptoms measured by t-scores > 65 based on the DSM-5 ADHD-oriented scales in ASR. Finally, the TDC group included those children who were absent in the ADHD-F and ADHD-NF groups. The detailed inclusion and exclusion steps are shown in Figure 1.

The short version of the Edinburgh handedness inventory was used to assess a handedness score of “right”, “both”, or “left” based on 4 questions about preferred hand using for writing, throwing, using a spoon, and using a toothbrush [60]. Due to lots of missing information in the ABCD Youth Pubertal Development Scale and Menstrual Cycle Survey History [61], the caregiver’s score has been used to calculate the Puberty Category Score (PCS). To assess PCS for boys, scores of body hair growth, voice change, and facial hair items were extracted from Pubertal Development Scale questionaries and then were summed and categorized as follows: 3 as pre-pubertal; 4 or 5 as early-pubertal; 6, 7, or 8 as mid-pubertal; 9, 10, or 11 as late-pubertal; 12 as post-pubertal. To assess PCS for girls, scores for two questions from Pubertal Development Scale questionaries including body hair growth and breast development were summed and categorized as follows using the menarche variables: 2 and no menarche as pre-pubertal; 3 and no menarche as early-pubertal; 3 and more than 3 and no menarche as mid-pubertal; 7 and less than 7 and menarche as late-pubertal; 8 and menarche as post-pubertal [62,63]. The current version of the parent-based Child Behavior Checklist (CBCL) was used to assess the behavioral problems. The CBCL is a multi-axial assessment (available in three versions including parent report, teacher report, and adolescent self-report) which has been developed as a general assessment instrument for child behavioral–emotional problems and competencies [59,64]. 

### 2.2. MRI Acquisition Protocol

The ABCD neuroimaging data acquisition protocols and parameters have been previously published in detail [65]. Briefly, three-dimensional T1-weighted structural MRI data were acquired on a 3T scanner (either Siemens, General Electric, or Philips) with a 32-channel head coil. The T1-weighted was acquired using inversion prepared RF-spoiled gradient echo pulse sequence with the following parameters: voxel size = 1.0 × 1.0 × 1.0 mm^3^; flip angle = 8°; repetition time/echo time for Siemens, Philips, and GE scanners were 2500/2.88, 6.31/2.9, and 2500/2, respectively; field of view = 256 × 256 mm^2^ for Siemens and GE scanners and 256 × 240 mm^2^ for Philips scanner. 

### 2.3. Individual-Level Structural MRI Data Analyses

All participants who had T1-weighted structural images with moderate to severe motion or visible artifacts were excluded from further analyses after careful visual inspection. Three-dimensional cortical reconstruction and segmentation for thickness and area estimations using T1-weighted data were performed utilizing Freesurfer version 7.2.0 (https://surfer.nmr.mgh.harvard.edu (accessed on 13 July 2021) [66,67]. Individual level data processing consists of multiple stages including skull stripping, volumetric labeling, intensity normalization, white matter segmentation, surface atlas registration, surface extraction, and gyral labeling. For each subject, the transformation matrix was computed using an affine registration method by registering each data point in Talairach space. Then, intensity normalization was performed using the Voronoi partitioning algorithm to avoid segmentation errors caused by fluctuations in intensity which resulted from magnetic field inhomogeneities. To remove the skull and other non-brain tissues, a brain mask was generated based on T1 volume using a deformable template algorithm. Then, these masks for each participant were inspected visually by two raters. Brain masks with minor problems (34 subjects), for example, the inclusion of partial dura matter, were edited. Subjects with major errors in the brain masks (30 subjects) were excluded from further analyses.

After applying brain masks, each voxel was categorized as white matter or non-white matter based on intensity and estimated bias field. Next, in the white surface formation stage, Freesurfer refined generated surfaces for hemispheres by following intensity gradients between white matter and gray matter. Then, at the pial surface formation, Freesurfer nudged white surface to follow the intensity gradients between gray matter and cerebrospinal fluid. The cortical thickness in each of 68 bilateral cortical regions was then estimated based on the distance between white and pial surfaces at each location. The border surface between gray matter and white matter was inflated into a sphere, and then surface-based labeling was used to segment both left and right hemispheres into different sub-regions in each of 68 bilateral cortical regions. Thirty-seven subcortical structures/nuclei regions’ volumetric segmentations were performed at several stages including affine registration, initial volumetric labeling, inhomogeneities correction, volumetric alignment to the MNI305 atlas, and final segmentation.

### 2.4. Group Statistical Analyses

SPSS (IBM Corp. Released 2020. Version 27.0. Armonk, NY, USA) was used to compare clinical, neurocognitive, and demographic measures of the participants between groups of controls and ADHD probands, as well as between the two ADHD subgroups (ADHD-F and ADHD-NF). Independent sample *t*-tests were used for continuous variables (displayed as the mean and standard deviation). Chi-square tests were used for categorical variables (displayed as number and frequencies).

Mixed model analysis of covariances (ANCOVA) was used to compare the structural T1 neuroimaging measures including regional cortical thickness and surface area of 68 cortical regions and volume of 37 subcortical structures/nuclei between the groups of TDC and ADHD probands and further between the subgroups of ADHD-F and ADHD-NF. Age and sex were used as covariates; we also added estimated total intracranial volume (ETIV) as covariate for volume and surface area because they scale with size of the head [68,69]. Bonferroni correction method [70] was used to adjust *p*-values and control potential false positive results of these group comparisons. Additionally, full model analyses were conducted (shown in Appendix A) to investigate other demographic effects including Sex, Age, Handedness, Race, Parents’ Education, Puberty, IQ, as well as MRI manufacturer as a random effect (ETIV has been used as covariate for volume and surface area).

Pearson correlation was performed to analyze whether the neuroanatomical brain measures that showed between-group differences were correlated with the clinical symptom measures (the raw scores for inattentive and hyperactivity–impulsivity symptoms score collected from KSADS-5 and DSM-oriented t-scores collected from CBCL). Again, Bonferroni correction method was used to adjust *p*-values and control potential false positive results of these comparisons.

## 3. Results

The final sample used in the main analysis comprised a total of 606 participants including 309, 165, and 132 in groups of TDC, ADHD-NF, and ADHD-F, respectively. Participants’ demographic information and corresponding group statistical comparisons are shown in Table 1. TDC and ADHD groups were matched on all demographic factors.

Group comparisons in neuroimaging measures showed that, compared to the TDC group, the ADHD probands had significantly decreased surface areas in the left cuneus (F= 4.31, *p* = 0.038) and significantly increased surface areas in the right middle temporal (F= 6.09, *p* = 0.028). There were no significant findings for cortical thickness or volume between groups of TDC and ADHD (Table 2).

Relative to children with ADHD-NF, the subgroup ADHD-F showed significantly decreased cortical thickness in the right pars orbitalis (F = 4.69, *p* = 0.031); significantly increased surface area in the left inferior temporal (F = 9.56, *p* = 0.008) and left middle temporal (F = 6.09, *p* = 0.024) lobes; significantly greater volume in the right thalamus (F = 6.43, *p* = 0.024) (Table 3).

Brain–behavioral correlation analyses demonstrated that among children with ADHD-F, a higher volume of the right thalamus was significantly correlated with a reduced t-score of ADHD problems from DSM-oriented CBCL scales (r^2^ = 0.039, *p* = 0.023) (Figure 2). No significant brain–behavioral correlations were observed in the ADHD-NF subgroup in this large community-based sample.

## 4. Discussion

Here, we present a highly powered study of cortical and subcortical structural brain differences between the subgroups of familial and non-familial of ADHD probands in children, as well as between ADHD and TDC groups. Compared to controls, the ADHD group had a significantly reduced surface area of the cuneus in the left hemisphere. The cuneus is one of the critical regions in processing visual information and has been suggested to be part of the default mode network [71,72,73]. Several neuroimaging studies have reported structural alterations in the cuneus in children with ADHD. In line with our results, one study has reported a reduced gray matter volume in the left cuneus in children with ADHD when compared to typically developing children [74]. Meanwhile, diffusion-weighted imaging studies have reported an increased radial diffusivity in the left cuneus in both subtypes of combined ADHD and inattentive ADHD compared to healthy control subjects, and the increased axial diffusivity in the right cuneus was associated with the diagnosis of ADHD [75,76]. In adults, task-based and resting-state fMRI studies also showed functional alterations in the cuneus [77,78]. In addition, our results showed that relative to the controls, the ADHD group showed an increased surface area in the right middle temporal gyrus. Brain morphology studies have reported gray matter volume and thickness alterations in the regions of the right temporal lobe in children with ADHD [79,80,81]. Diffusion-weighted imaging studies have also reported fractional anisotropy (FA) abnormalities in the major white matter tracts that involve the right temporal lobe [82,83,84]. Together with findings of existing imaging studies, our study in this large and community-based sample of subjects from the ABCD study suggests that structural alterations in the cuneus and temporal gray matter may play a role in the onset of ADHD symptoms in school-aged children.

Within the probands, our findings showed significantly increased gray matter volume of the right thalamus in the ADHD-F group compared to the ADHD-NF group. Furthermore, the increased volume of the right thalamus was significantly correlated with the reduced t-score of ADHD problems from DSM-oriented CBCL scales. Thalamus is one of the critical components of the Cortico-Striato-Thalamo-Cortical (CSTC) loops which subserve attention and cognitive processing [47,85,86]. It has been shown that the pathogenesis of ADHD is theorized to be involved in anatomical and functional disruptions in CSTC loops [48,87,88]. Investigations in thalamus-related structures and circuits studies in the fMRI study have resulted in significantly decreased regional activation in the bilateral thalami (in pulvinar nuclei) and significant elevation in the connectivity between the right pulvinar and bilateral occipital areas in children with ADHD [88]. Similarly, a DTI study observed decreased white matter connectivity between the thalamus and striatum in children [44]. A large number of structural neuroimaging studies from our group and others have reported thalamus or thalamic-related structural alterations with widely inconsistent findings [44,89,90,91]. For instance, Rosch et al. [89] and Xia et al. [44] reported a significantly reduced volume of the thalamus among children with ADHD compared to controls. Mooney et al. [92] and Fu et al. [90] reported a significantly larger gray matter volume in the ADHD and the inattentive ADHD groups, respectively. One recent multi-site study reported no significance in the volume of the thalamus between children with ADHD and controls [93]. Compared to typically developmental controls, Chen et al. [91] reported significantly reduced gray matter in the volume of the thalamus in monozygotic twins discordant for ADHD. In addition, it has been shown that the volume of the thalamus was significantly positively correlated with the severity of ADHD symptoms [90]. These relatively inconsistent findings are likely due to methodological differences such as different methods of data acquisition and analyses, limited sample sizes, and sample related biases, such as involving samples from heterogeneous etiological sources. Combined with the existing study, our results suggest that structural anomalies in the thalamus may have a strong heredity origin (with biologically distinct representation of ADHD) and play a critical role in children with ADHD and high family risk factors. These findings may serve as a potential biomarker for treatments and important interventions in children with ADHD-F for future clinical studies and practice.

Furthermore, within the ADHD probands, we found that children with ADHD-F had significantly thinner cortical thicknesses in the right pars orbitalis compared to children with ADHD-NF. Right pars orbitalis, as a part of the right inferior frontal gyrus (rIFG), plays a key role in behavioral responses inhibition [94,95]. It has been shown that children with ADHD have deficient inhibitory control compared with typically developing controls [96,97,98]. Ducharme et al. [99] have shown significant decreased association between the thinning rate of cortical thickness in rIFG, as well as an abnormal growth pattern of rIFG, and inattention symptoms in children. Another study has reported shared neuroanatomical alterations in the gray matter volume in rIFG between ADHD probands and their unaffected siblings [40]. A recently published fMRI-neurofeedback study in adolescents with ADHD has shown an association between the right frontal inferior cortex and clinical symptoms of ADHD in the fronto-striatal and default mode network regions [100]. Together with existing investigations, we suggest that ADHD-F may have significantly more severe neuroanatomical underdevelopment in rIFG than ADHD-NF, which reflects the anatomical heterogeneity of this group in terms of family risk factors. Future longitudinal studies may help to find whether such childhood structural alterations in the frontal lobe in ADHD-F are associated with persistence/remission of the ADHD symptoms in adulthood.

## 5. Limitations and Future Directions

The present study has a few limitations. One limitation of the study is that the ABCD study only included the ADHD assessment for one of the parents. To minimize the confounding effects in the ADHD-NF group, we further excluded subjects that had any potentially ADHD-related family histories based on the collected family history questions. Another limitation is that the parametric statistical models used in this study have a limited capacity to investigate multiple variables and their interactions simultaneously. Future studies using machine learning methods may effectively extract the high-dimensional information and translate the complex neuroimaging patterns into clinical relevance. Finally, the cross-sectional study design of this study only focused on understanding the anatomical alterations and their interactions with the ADHD symptoms in children with ADHD-F vs. ADHD-NF at 9–10 years of age. Future research with a longitudinal study design will facilitate the understanding of how these common and distinct neuroanatomical features in children with ADHD-F vs. ADHD-NF grow along with age and whether the ADHD-F-specific childhood neural markers (associated with the thalamus and inferior frontal regions) in affected individuals have an important predictive value for future clinical outcomes associated with the persistence/remission of ADHD symptoms.

## 6. Conclusions

In conclusion, the current study found that relative to group-matched controls, school-aged children with both ADHD-F and ADHD-NF had common gray matter structural abnormalities associated with the cuneus and temporal lobe. Whereas, within the ADHD probands, children with ADHD-F showed a significantly increased volume of the right thalamus and decreased cortical thickness of the right IFG when compared to ADHD-NF; the volume of the right thalamus showed significant negative correlation with the severity of ADHD symptoms (scored using the DSM-5-oriented scales in CBCL) in children with ADHD-F, but not in those with ADHD-NF. These findings resulted from analyses of a big sample of children from community-based recruitment. Therefore, findings reported in our study are statistically much more powerful than those reported in studies with small sample sizes. Meanwhile, this is the first study in the field investigating neuroanatomical substrates of ADHD-F vs. ADHD-NF in two large independent subgroups of ADHD in children. The results suggest that ADHD-F may represent a biologically more homogeneous subgroup of ADHD. These findings have the potential in guiding future studies with longitudinal designs to understand the developmental trajectories of these ADHD-F-specific neural markers and their roles in clinical outcomes associated with the persistence/remission of ADHD symptoms in affected individuals.

## Figures and Tables

**Figure 1 brainsci-13-00046-f001:**
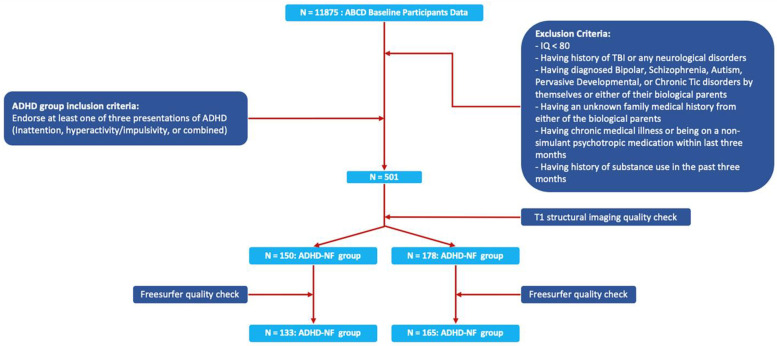
ADHD participants’ inclusion and exclusion flowchart.

**Figure 2 brainsci-13-00046-f002:**
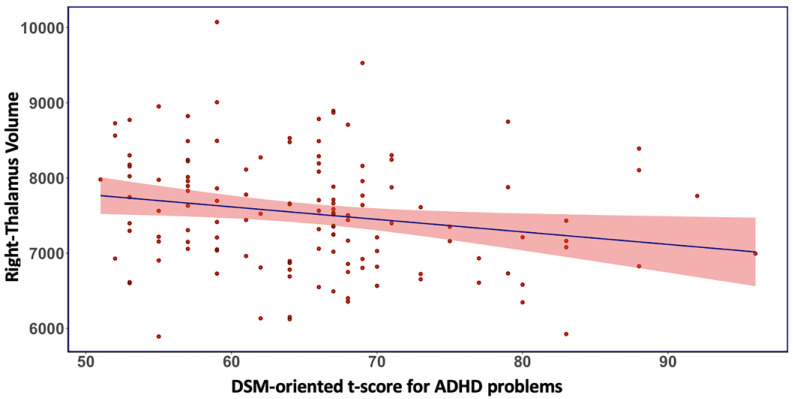
In the familial proband of ADHD, greater volume of the right Thalamus was significantly associated with reduced DSM-oriented t-score for ADHD problems from the CBCL scales.

**Table 1 brainsci-13-00046-t001:** Demographic and clinical characteristics comparisons between groups of controls and ADHD probands, and between ADHD-F and ADHD-NF subgroups.

	TDC (N = 309) Mean (SD) or N (%)	ADHD (N = 297) Mean (SD) or N (%)	*p*-Value	ADHD-NF (N = 165) Mean (SD) or N (%)	ADHD-F (N = 132) Mean (SD) or N (%)	*p*-Value
**Age** (months)	119.62 (7.26)	118.93 (7.64)	0.256	118.28 (7.64)	119.75 (7.59)	0.099
**Sex:**			0.308			0.296
Female	128 (41.4)	111 (37.4)		66 (40.00)	45 (34.09)	
Male	181 (58.6)	186 (62.6)		99 (60.00)	87 (65.90)	
**Handedness:**			0.436			0.679
Right-Handed	246 (77.1)	229 (77.1)		130 (78.78)	99 (75.00)	
Left-Handed	23 (7.4)	19 (6.4)		9 (5.45)	10 (7.57)	
Both-Handed	40 (12.9)	49 (16.5)		26 (15.75)	23 (17.42)	
**Puberty Category Score:**			0.417			0.383
Pre-Pubertal	185 (61.7)	169 (58.5)		93 (58.9)	76 (58.0)	
Early-Pubertal	67 (22.3)	68 (23.5)		33 (20.9)	35 (26.7)	
Mid-Pubertal	41 (13.7)	49 (17.0)		31 (19.6)	18 (13.7)	
Late-Pubertal	7 (2.3)	3 (1.0)		1 (0.6)	2 (1.5)	
**IQ** (Picture Vocabulary)	107.03 (16.51)	107.23 (18.54)	0.889	106.07 (18.17)	108.59 (19.23)	0.253
**Race:**			0.457			0.718
Caucasian	221 (71.5)	214 (72.05)		120 (71.21)	94 (72.72)	
African-American	36 (11.7)	36 (12.12)		22 (13.33)	14 (10.60)	
More than one race	25 (8.1)	30 (10.10)		15 (9.09)	15 (11.36)	
Other Races	27 (8.7)	17 (5.72)		8 (4.84)	9 (6.81)	
**Income ^1^:**			0.954			0.002
<USD 50,000	66 (22.8)	60 (22.3)		25 (16.7)	35 (29.4)	
USD 50000-USD 100,000	90 (31.1)	87 (32.3)		43 (28.7)	44 (37.0)	
>USD 100,000	133 (46)	122 (45.4)		82 (54.7)	40 (33.6)	
**Parental Education:**			0.293			0.794
No high school diploma	21 (6.8)	15(5.1)		10 (6.1)	5 (3.8)	
High school diploma	30 (9.7)	17 (5.7)		10 (6.1)	7 (5.3)	
Some College	96 (31.1)	92 (31)		48 (29.1)	44 (33.3)	
Bachelor’s degree	80 (25.9)	89 (30		52 (31.5)	37 (28.0)	
Graduate degree	82 (26.5)	84 (28.3)		45 (27.3)	39 (29.5)	

^1^ The incomes were measured by USD per year. TDC: typically developed children; ADHD: attention-deficit/hyperactivity disorder; ADHD-F: familial ADHD; ADHD-NF: non-familial ADHD; SD: standard deviation.

**Table 2 brainsci-13-00046-t002:** Gray matter neuroimaging measures that show significant differences between control and ADHD groups with age, sex, and ETIV (for volume and area) as covariates.

Anatomical Location	Measure	TDC Mean ± SD	ADHD Mean ± SD	F-Value	*p*-Value after Bonferroni Correction
L. Cuneus	Surface Area	1676.39 ± 218.33	1633.97 ± 229.09	4.31	0.038
R. Middle Temporal	Surface Area	3997.17 ± 502.91	4011.51 ± 510.05	6.09	0.028

TDC: typically developed children; ADHD: attention-deficit/hyperactivity disorder; ETIV: estimated total intracranial volume; SD: standard deviation; L: left hemisphere; R: right hemisphere.

**Table 3 brainsci-13-00046-t003:** Gray matter neuroimaging measures that show significant differences between Familial and Non-Familial groups with age, sex, and ETIV (for volume and area) as covariates.

Anatomical Location	Measure	ADHD-NF Mean ± SD	ADHD-F Mean ± SD	F-Value	*p*-Value after Bonferroni Correction
R. Pars Orbitalis	Cortical Thickness	3.00 ± 0.17	2.95 ± 0.15	4.69	0.031
L. Inferior Temporal	Surface Area	3711.17 ± 498.36	3905.69 ± 579.72	9.56	0.008
L. Middle Temporal	Surface Area	3579.24 ± 446.97	3730.73 ± 540.07	7.52	0.024
R. Thalamus	Volume	7325.65 ± 649.59	7526.62 ± 770.62	6.43	0.024

ADHD: attention-deficit/hyperactivity disorder; ADHD-F: familial ADHD; ADHD-NF: non-familial ADHD; ETIV: estimated total intracranial volume; SD: standard deviation; L: left hemisphere; R: right hemisphere.

## Data Availability

Data are available upon reasonable request to the corresponding author.

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
