# Peer review of "Distinct Thalamic and Frontal Neuroanatomical Substrates in Children with Familial vs. Non-Familial Attention-Deficit/Hyperactivity Disorder (ADHD)"

_brainsci, 2022, doi:10.3390/brainsci13010046_

Round 1
Reviewer 1 Report
The authors present a manuscript to investigate the neuroanatomical differences between the children with ADHD having or not having positive family risk factor. The manuscript is well-organized, however, it would be great if authors can highlight the following concerns:
1. The group comparisons were controlled for only age, sex, and eTIV. It would be great if authors can conduct additional analysis of the effects of other demographic factors, e.g., socioeconomic status, race.
2. The paper showed that the data were collected from Siemens, Philips and GE, did authors conduct any analysis to exclude the batch effect? How did authors remove those effects?
3. Based on the results, the temporal lobe also showed group differences between controls and ADHD, and between ADHD-F and ADHD-NF. What are authors' thought on the results in temporal lobe?
Reviewer 2 Report
15 November 2022
Manuscript ID: brainsci- 2061568
Type: Article
Title: ‘Neuroanatomical Substrates Associated with Childhood Familial vs. Non-familial Attention-Deficit/Hyperactivity Disorder (ADHD)’ by Baboli R et al., submitted to Brain Sciences
Dear Authors,
The present research article by Baboli and colleagues, entitled ‘Neuroanatomical Substrates Associated with Childhood Familial vs. Non-familial Attention-Deficit/Hyperactivity Disorder (ADHD)’ is a well-written and useful summary on the status of knowledge of the influence of having or not having positive family risk factor in neuroanatomy of the brain of children with ADHD. For this purpose, cortical thickness, surface area and volume-based measures were extracted and compared in a total of 606 participants, including individuals with familial ADHD (ADHD-F), non-familial ADHD (ADHD-NF), and typically developed children. Results showed that compared to controls, ADHD pro-bands showed significantly reduced grey matter surface area in the left cuneus and that ADHD-F sub-group showed significantly increased grey matter volume in the right thalamus and significantly thinner cortical thickness in the right pars orbitalis, when comparing with the ADHD-NF. Also, among ADHD-F, increased volume for the right thalamus was significantly correlated with reduced ADHD problems.
The main strength of this manuscript is that it addresses an interesting and timely question, describing how children with ADHD family history show significantly increased volume in the thalamus and decreased cortical thickness in the right inferior frontal gyrus. In general, I think the idea of this article is really interesting and the authors’ fascinating observations on this timely topic may be of interest to the readers of Brain Sciences. However, some comments, as well as some crucial evidence that should be included to support the author’s argumentation, needed to be addressed to improve the quality of the manuscript, its adequacy, and its readability prior to the publication in the present form, in particular reshaping parts of the Introduction and Methods sections by adding more evidence and theoretical constructs.
Please consider the following comments:
1. I suggest changing the title. In my opinion, in the present form it seems to be too wordy and not enough informative and appropriate.
2. Abstract: According to the Journal’s guidelines, the abstract should be a total of about 200 words maximum. Please correct the actual one. Also, I recommend proportionally presenting the background, the objectives, the methods, and the conclusion. The background should include the general one, the detailed, and the current issue addressed to this study. The conclusion should clarify the potential and the advance this study has provided in the fields.
3. Keywords: Please list ten keywords and use them as many as possible in the first two sentences of the abstract.
4. A graphical abstract that will visually summarize the main findings of the manuscript is highly recommended.
5. Introduction: The ‘Introduction’ section is well-written and nicely presented, with a good balance of descriptive text and information about epidemiology and characteristics of ADHD. Even though the authors decided to take a narrow view of etiology and neurobiological markers of ADHD disorder, I believe that more information about pathophysiology, more specifically about frontolimbic dysfunction involved in altered inhibitory control associated with this disorder, would provide a better and more accurate background. Thus, I suggest the authors to make such effort to provide a brief overview of the pertinent published on brain dysfunctions in ADHD. In this regard, I believe focusing on links between frontal and parietal abnormalities and behavioral features of this disorder (https://doi.org/10.3389/fnbeh.2022.946263; https://doi.org/10.3389/fnbeh.2022.998714). Secondary, the authors may also consider adding evidence that target pathomechanisms of neurodevelopmental disorders, searching for novel targets, and developing new neuroprotective agents against these psychiatric diseases (https://doi.org/10.1007/s00702-022-02513-5; https://doi.org/10.3390/biomedicines9070734).
6. Participants: Data on participants and information about clinical assessment for patients’ selection are not adequately explained. For this reason, I would ask the authors to specify how they estimated the exact number of participants and provide more information on the diagnostic tests used for clinical evaluation.
7. Participants: Could the authors better describe and provide reference for the tests utilized in this experiment?
8. Individual-level structural MRI data analyses: In my opinion, this section is well organized, but it illustrates findings in an excessively broad way, without really providing full statistical details, to ensure in-depth understanding and replicability of the findings. I suggest rewriting this section more accurately, and to present statistical data not only in the main text, but also in the tables.
9. Although not mandatory, I believe that a proper ‘Conclusions’ paragraph, in which authors could provide some thoughtful as well as in-depth considerations, would be very useful to present the take-home message as the experts of this field. The authors should make their effort to explain the theoretical implication as well as the translational application of their research.
10. In according to the previous comment, I would ask the authors to include a proper ‘Limitations and future directions’ section before the end of the manuscript, in which authors can describe in detail and report all the technical issues brought to the surface.
11. Figures: All figures should be presented in color.
12. References: The authors should consider revising the bibliography, as there are several incorrect citations. Indeed, according to the Journal’s guidelines (https://www.mdpi.com/journal/brainsci/instructions), they should provide the abbreviated journal name in italics, the year of publication in bold, the volume number in italics, and a period after the page number for all the references.
Overall, the manuscript contains 3 tables, 2 figures and 76 references. I believe that the manuscript might carry important value in describing how children with ADHD family history show significantly increased volume in the thalamus and decreased cortical thickness in the right inferior frontal gyrus. I hope that, after these careful revisions, this paper can meet the Journal’s high standards for publication. I am available for a new round of revision of this paper.
I declare no conflict of interest regarding this manuscript.
Best regards,
Reviewer
Round 2
Reviewer 2 Report
13 December 2022
Manuscript ID: brainsci-2061568
Type: Article
Title: ‘Neuroanatomical Substrates Associated with Childhood Familial vs. Non-familial Attention-Deficit/Hyperactivity Disorder (ADHD)’ by Baboli R et al., submitted to Brain Sciences
Dear Authors,
I am pleased to see that the authors took my comments seriously and solved almost all issues I raised in the previous round of the peer-review session. The manuscript entitled ‘Distinct Thalamic and Frontal Neuroanatomical Substrates in Children with Familial vs. Non-familial ADHD’ is a well-written and nicely presented research article on the status of knowledge of the influence of having or not having positive family risk factor in neuroanatomy of the brain of children with ADHD. I just noticed two minor points I would like the authors to correct to finalize my peer-review session.
Comments:
1. Please expand abbreviation in the title.
2. Please place a period (.) instead of a comma (,) in front of doi. number.
I believe that the manuscript carries important value in describing how children with attention deficit hyperactivity disorder family history show significantly increased volume in the thalamus and decreased cortical thickness in the right inferior frontal gyrus. This manuscript now meets the Journal’s high standards for publication. I am looking forward to see a manuscript written by the authors in near future.
I declare no conflict of interest regarding this manuscript.
Best regards,
Reviewer
